# The Use of Robotic Technology in the Healthcare of People above the Age of 65—A Systematic Review

**DOI:** 10.3390/healthcare11060904

**Published:** 2023-03-21

**Authors:** Ann-Chatrin Linqvist Leonardsen, Camilla Hardeland, Ann Karin Helgesen, Carina Bååth, Lilliana del Busso, Vigdis Abrahamsen Grøndahl

**Affiliations:** 1Faculty of Health, Welfare and Organization, Ostfold University College, Postal Box Code 700, 1757 Halden, Norway; 2Department of Anesthesia, Ostfold Hospital Trust, Postal Box Code 300, 1714 Grålum, Norway; 3Department of Health Sciences, Faculty of Health, Science, and Technology, Karlstad University, SE-651 88 Karlstad, Sweden

**Keywords:** animal-like, human-like, older people, robot, systematic review, technology

## Abstract

Aim: The integration of robots can help provide solutions in regards to the need for an increase in resources in healthcare. The aim of this review was to identify how robots are utilized in the healthcare of people who are over the age of 65 and how this population experiences interacting with healthcare robots. Design: A systematic literature review with an integrated design was conducted. Methods: A literature search was performed in the electronic databases CINAHL via EBSCO, EMBASE, and Medline via Ovid. Content analysis was performed to assess the studies that were included in this review. Results: A total of 14 articles were included. Participants in the studies included 453 older people ranging from 65 to 108 years of age. Nine of the studies focused on people with dementia or cognitive impairment. Seven studies included different types of socially assistive human-like robots, six of the studies included two different types of animal-like robots, and one study focused on a robotic rollator. The robots mainly served as social assistive- or engagement robots.

## 1. Background

In modern societies, the average life expectancy has increased along with the demand for human resources for elderly care [1]. Today, staff shortages in general, and shortages in nursing staff in particular, as well as high turnover rates, present major challenges to healthcare delivery [2]. The demand for efficiency and everyday experiences of stress and feeling overwhelmed among nursing staff makes providing person-centered and dignified care challenging [3]. The staffing needs within healthcare services are estimated to double by 2060 [4]; therefore, new and innovative solutions are necessary.

Care personnel accounts for a significant proportion of healthcare costs [5,6], and the robotization of the work for care personnel may therefore have significant economic benefits. As such, the integration of robots into healthcare may be a response to the need for increased human resources in these services [7,8,9]. Internationally, different types of robots have been introduced. These have been categorized into companion robots, telepresence robots, manipulator service robots, rehabilitation robots, health monitoring robots, reminder robots, domestic robots, entertainment robots and fall detection/prevention robots [10]. Several areas of use for robots in elderly care were identified, including, for example, affective therapy, cognitive training, social facilitation, companionship, physiological therapy, household tasks, personal care tasks, companionship tasks, and communication tasks [11,12]. Moreover, studies have explored older people’s perceptions of robots, showing a diverse and complex picture [11,13,14], prompting a need for further research.

The majority of research conducted in this field focused on different stakeholders’ perspectives on the technology itself or on the roles of robots in specific settings. Our aim in this review was to identify the use of robots in the healthcare of people over the age of 65, regardless of focus, context, type of robot or diagnosis. More specifically, we wanted to explore the different types of robots used and how these were utilized and experienced by the participants. The findings from this review thus contributed to a better understanding of what is needed to develop, implement and improve healthcare services through the inclusion of robotic technologies.

## 2. Materials and Methods

We conducted a systematic literature review with an integrated design, including findings from qualitative, quantitative and mixed methods studies [15]. 

Initially, we conducted a search in Cochrane Systematic Reviews to assess whether similar reviews had been published. We only identified five review protocols, which all focused on specific conditions or settings. Consequently, a systematic literature search was performed in collaboration with a specialist librarian in the electronic databases CINAHL via EBSCO, EMBASE, and Medline via Ovid in the period 5 July to 30 August 2021. End-Note X8 and Rayyan QCRI software were used to handle the references [16]. To structure the review, we used the Preferred Reporting Items for Systematic Reviews and Meta-Analyses (PRISMA) guidelines [17].

### 2.1. Search Strategies

The search strategies applied in this review were developed based on the PICO (patient-intervention-comparison-outcome) framework [18] (see Table 1).

Keywords, Mesh-terms and free text terms were then used separately and in combination with the boolean operators *OR* or *AND*, and combined with a truncation if applicable.

Table 2 gives an example of a search strategy in Medline. Searches in the other databases were similar to the search strategy in Medline, using the same terms and phrases, as well as Boolean operators OR/AND.

### 2.2. Inclusion Criteria


Participants 65 years or above (in line with the World Health Organization’s definition of ‘older people’ [19]);Use of robotic technology;Scandinavian or English language;Peer-reviewed;Not limited to study design or methodological approach;Published from 1 January 2011–30 August 2021.


### 2.3. Exclusion Criteria


Surgical procedures;Product development;Reviews;Conference abstracts;Abstracts;Errata;Letter to editors;Unpublished material;Dissertations;Management systems, such as electronic health records that allow the acquisition, transmission and storage of patient data;Computerized decision support systems, including diagnostic support.


### 2.4. Study Selection

The standard PRISMA flowchart was used to provide the process of study selection; please see Figure 1, which gives an overview of the identification, screening and selection process. The electronic database searches identified 1224 articles. After the removal of duplicates, 1017 articles remained. Furthermore, 422 articles were not published in Scandinavian or English language/non-scientific papers/surgical procedures and were consequently excluded. After this initial exclusion process, which was conducted by five of the authors, the remaining 595 article titles and abstracts were screened against the inclusion criteria by 2 of the authors, resulting in an additional 453 articles being excluded. The remaining 142 articles were read in full text, and an additional 90 articles were excluded for not meeting the inclusion criteria. The majority of these articles were studies in which participants were not limited to people above the age of 65 years. Potential conflicts were discussed with a third author until an agreement was achieved. This study selection process resulted in 52 articles remaining to be assessed for quality. 

### 2.5. Quality Appraisal 

The Critical Appraisal Skills Programme checklists (CASP) [20] were used to appraise the quality of the identified articles. We used checklists for randomized controlled trials, qualitative studies, case-control studies and cohort studies adhering to the respective study design. For mixed-methods studies, these were combined as well. Firstly, the appraisals were conducted by four authors independently. We used a simple scoring system where ‘criterion is completely met’ = 2, ‘criterion is partially met’ = 1, and ‘criterion not applicable, not met, or not mentioned’ = 0. A total score of 20 = was interpreted as high quality, 16–19 as moderate quality, and ≤15 as low quality. Secondly, the results from the quality appraisals were compared and discussed between all four authors, and after agreement, articles with medium and high-quality ratings were included. A total of 38 articles were excluded due to low quality. Any differences or uncertainties were discussed by all authors until an agreement was reached [21]. Ultimately, a total of 14 articles were included in this review. 

### 2.6. Data Extraction and Synthesis

We used a standardized data extraction template that included the first author, year, study country, title, aim, study design and ethics, robotic technology, setting, the sample, including participants’ characteristics, and a summary of relevant findings. This extraction was conducted by 4 of the authors. Then, all authors assessed the template and agreed on a final version. 

Data were analyzed using Hsieh and Shannon’s [21] conventional content analysis, which is considered an appropriate method of analysis in descriptive studies. In the first phase of the analysis, the first author familiarized herself with the data by reading and re-reading the results sections of the included articles. The data was then coded word by word in order to identify the key concepts utilized in the articles. In the next phase, the first author reviewed the codes across all 14 articles in order to confirm and/or refuse the codes identified. The codes were then collated into subcategories, and the main categories were then identified based on how the different codes were related and linked. Finally, categories and subcategories were discussed in an iterative process between all of the authors until a consensus was reached.

### 2.7. Systematic Review Registration Number

Systematic Review Registration Number: 290621 PROSPERO, an international database of prospectively registered systematic reviews in health and social care developed and managed by the Centre for Reviews and Dissemination (CRD) at the University of York and funded by the UK’s National Institute for Health Research (NIHR).

## 3. Results

### 3.1. Study Selection

In total, 1224 records were identified, of which 595 records were screened by focusing on title and abstract only. Of these, 53 full-text records were screened and quality assessed, leading to a total of 14 publications being included in the review. The included studies represented research conducted during the period from 2012–2021 and were performed in ten different countries. The studies included a total of 453 elderly people, with the age of participants ranging from 65 to 108 years. It was not possible to calculate the mean or median age due to some studies not reporting all ages. In addition, not all studies reported the distribution of gender. However, in studies that reported gender (n = 10), 60 percent of participants were female. In addition, two studies included a total of 14 family members, and three studies included a total of 37 staff members/professional caregivers. Nine of the studies included people with dementia or cognitive impairment. Seven studies focused on different types of socially assistive humanlike robots, while four of the studies focused on the robotic seal Paro, and one study compared Paro and a human-like robot. Additionally, one study focused on a robotic cat, and one study focused on a technological device used for physiological therapy.

In total, three themes were identified: (1) the use of and interaction with human-like robots, (2) the use and experience of animal-like robots; and (3) robotic devices in the care of older people.

### 3.2. Use of and Interaction with Human-like Robots

The human-like robots varied in presentation, size, and possibilities/activities. The robots were: Kabochan [21], Sophie and Jack [22], Guide [23], HIRO [24], the Ro-tri [25], NAO [26], Betty [27], and Pepper [28] (see Appendix A for more information about the robots). Five studies focused on older persons with dementia [21,22,23,27,28], and one study divided participants into ‘no cognitive impairment’ (n = 8), ‘mild cognitive impairment’ (n = 5) or ‘moderate cognitive impairment’ (n = 2) [26], and in one study 11 of 14 participants had either mild cognitive impairment (n = 10) or Alzheimer’s disease (n = 1) [25].

Four studies focused on observing older peoples’ actions when interacting with a robot. The robot Kabochan was mainly used for cuddling (13.6%), talking with (8.7%), coaxing to sleep (4.7%), moving feet and arms (3.5%), carrying (2.4%), tidying appearance (1.5%), tickling foot (1.4%), and feeding the robot (0.2%) [21]. However, this particular robot did not lead to any changes in attitudes towards technology, perceived usefulness, technology anxiety or self-efficacy compared to the usual care. Sumioka et al. [24] observed older patients’ caring behaviors towards HIRO, which included talking and singing to it as well as caressing, hugging and rocking the robot. In addition, the researchers compared HIRO with a face and HIRO without a face. They found that significantly more older people in the ‘with face’ group (81.8%) continued to hold the robot during a five-minute period, than older people in the ‘without face’ group (60%). Moreover, six participants rejected the robot regardless of face/no-face. Fan et al. [25] found that older people were interested in and accepted the Ro-Tri robot. This was indicated by participants spending 77.7% of the time looking at the robot and 2.3% of the time looking toward their partners. This interest was maintained for a period of three weeks. Further, Khosla et al. [27] performed observations (n = 2043) of older people with dementia interacting with Betty through video recording through the robot in their homes over a 3 month period. This robot was used to encourage older people to engage in an activity such as singing and dancing (1014 observations), answering a quiz (321 observations), checking the weather report (256 observations), reading the news (161 observations), reading a book (124 observations), checking a calendar reminder (110 observations), and making a phone call (57 observations).

Fields et al. [26] explored older people exposed to the robot NAO and their scores of loneliness, depression, and face scores across six time periods. These measures had a significant decrease but differed between people with dementia and those without dementia. The degree of difference/change was slightly greater in participants without dementia. Papadopoulos et al. [28] focused on the robot Pepper and older peoples’ scores on the SF-36 mental health subscale when exposed to the robot. They found that older people’s emotional status improved slightly due to interacting with the robot. The robots Sophie and Jack [23] were shown to improve the capacity of caregivers towards older people through engaging in activities such as playing games or music.

The Guide robot was thought to be unsuitable for people with dementia. This was related to the fact that it was difficult to remember how to use it [23].

### 3.3. Use and Experience of Animal-like Robots

In total, five studies explored the effect of the robotic seal Paro on older patients. Of these, one study explored the effect of Paro on older patients with depression [29], two studies explored the effect of Paro on older people with dementia [23,30], one on older people with successive cognitive impairment [31], while in one study, participants had no cognitive impairment [32]. The studies varied between being exposed to Paro for ten minutes (with blood pressure taken before, during and after the exposure) [32] to having PARO present 24 hours a day, seven days a week for eight weeks [29]. Robinson et al. [23] compared Paro to the robot Guide. The comparisons included ‘touching,’ ‘talking to,’ and ‘being enthusiastic about,’ and the authors found that more older patients touched (100% versus 40%), talked to (60% versus 20%) and were enthusiastic about Paro in comparison to Guide (non-significant findings). Examples of positive comments exclusively used to describe Paro were: ‘beautiful looking’ and ‘had especially lovely eyes,’ ‘was lifelike,’ and ‘tactile.’

All the studies reported positive psychological effects of Paro, either on depression, loneliness or quality of life [29], providing comfort and relaxation and a distraction from the experience of pain [23,30], a feeling of bonding, a companionship [23,31] or «a friend to play with» [30]. One study described Paro as “a stimulus, which would keep people entertained” and that it “interacts with residents like an animal but will not try to escape the resident like a real animal might” [23]. Limitations of Paro were related to its voice, weight and programming, and that it was unable to walk [30].

In two of the studies, participants who did not show any interest in Paro were excluded from further analyses. One study excluded participants who did not show any interest in Paro and then identified a significant decrease in systolic/diastolic blood pressure and average heart rate over time in patients being exposed to Paro [32]. The other study excluded six of 20 patients who “refused to play with the robot”. Participants who did not wish to engage with the robot were of two types: (i) those who thought the robot was a silly toy and did not wish to play with it (ii) residents, both female) and those who only engaged with the robot when asked but who would rather ignore it [31].

Gustafsson et al. [33] found that the social robot cat JustoCat increased well-being, reduced loneliness, gave a sense of stability, stimulated participants to participate in other activities and increased their activity levels. Similarly to studies that included Paro, participants emphasized the cats’ nice faces, big eyes, and natural size and weight. The professional caregivers and relatives experienced the robotic cat as encouraging interaction and communication with older people with dementia. Professional caregivers and relatives also experienced the JustoCat as “something else to think about” and as discouraging repetitive behavior.

### 3.4. Robotic Device(s) in the Care of Older People

Only one study focused on a technological device, a robotic rollator, used for physiological therapy in older people [34]. The study explored whether a robotic rollator that provides navigation assistance in frail older adults with and without cognitive impairment improved navigation within a real-life environment in the intended user group. Findings showed that almost all participants were able to reach the destinations of the navigation path, independent of the navigation assistance provided by the robotic rollator or their cognitive status. 

## 4. Discussion

The findings of this review suggest that robotic technology is mainly utilized in terms of social assistive or engagement robots in the healthcare of people over the age of 65. We only identified one “technological device” [10], while the remaining research focused on companion or entertainment robots [10]. Moreover, nine of the 14 included studies focused on older people with dementia or cognitive impairment. Thus, this systematic review indicates that there is a knowledge gap regarding robotic technology in healthcare services provided to older people over the age of 65 years in general, as well as with regard to how robotic technology may contribute to finding a solution to the lack of resources in healthcare services. 

This systematic review suggests that robots, in the included studies, were mainly used in order to encourage older people to engage in activities such as singing and dancing, answering a quiz, checking the weather report, reading the news, reading a book and making a phone call. In addition, one study explored the use of a robotic rollator walker [10]. Nevertheless, few of the studies included in this review explored robotic technologies actually in use in the care of older people. Rather, the majority of the studies identified reactions and perspectives at one point in time, indicating that the technology was not implemented in older patients’ everyday lives over time. Furthermore, a systematic review conducted in 2018 explored how robotic technology can help older people in their daily lives, for instance, in relation to health monitoring, social isolation and dependent living [35]. As such, the authors identified nine different types of robots addressing challenges in aged care, including companion, manipulator service, telepresence, rehabilitation, entertainment, and domestic and fall detection/prevention robots. In contrast to the current review, which focused on people over the age of 65, Shishegar et al. [35] included studies of older adults or seniors. Hence, it can be argued that in order to gain a better understanding of the use and benefits and challenges of utilizing robotic technology in the daily care of people over the age of 65, future research should utilize longitudinal methods in order to explore the use and experiences of such technology with the same participants over time.

Previous research has identified positive cardiovascular effects of interacting with animals in older people [35,36]. Five of the studies included in this review focused on the robotic seal Paro and reported positive psychological effects such as relaxation and comfort [23,30,31,32]. In addition, Gustafsson et al. [33] found that the robotic cat JustoCat increased well-being and increased participants’ activity levels. Nevertheless, Hung et al. [37] conducted a scoping review of publications about Paro on persons with dementia and concluded that first-person perspectives of patients’ experiences and clinical needs were lacking. Moreover, they identified that few studies investigated processes of using robots effectively in different situations in order to meet clinical needs. In line with the current review, this indicated a need to explore contextualized first-person accounts of using animal-like robots in relation to the implementation and development of robotic technologies for elderly care.

Seven of the included studies focused on human-like robots [21,22,23,24,25,26,27,28]. In a systematic review of socially assistive robots in aged care, Vandemeulebroucke et al. [14] found that there was variability in preferences with regard to the appearance of socially assistive robots. As such, individual studies found a preference for machine-like appearances, human-looking robots or a combination of the two. The robots identified in the current review varied from resembling a 3-year-old boy, having a babyface to having no face at all and being from 1.6 m to approximately 15 inches in height. Vandemeulebroucke et al. [13,14] stated that the possible implementation and use of socially assistive robots are complicated due to their deep integration into social reality and that older adults should be able to choose the appearance of their socially assistive robot. Hence, they concluded that the implementation of socially assistive robots called for a broader conceptual analysis that focused on the possible relationship between the robots and their prospective users.

The research field, aiming to face similar purposes to the current review, is growing. For example, Fiorini et al. [38] interviewed 20 older people and 34 caregivers in Italy and the Netherlands regarding their needs concerning the personal mobility domains and their attitudes toward assistive robots. The results focused on the robots’ abilities and were identified as significant for computer developers and healthcare personnel. Moreover, Fiorini et al. [39] designed and developed a robotic sensorized handle for monitoring older adults’ grasping force. Moreover, Coradeschi et al. [40], as early as 2013, developed and tested a network of home sensors that could be automatically configured to collect data for a range of monitoring services, among these a semi-autonomous telepresence robot called GiraffPlus. 

New technologies might influence what we understand in terms of dignity, autonomy, reality, and social relations [41]. For example, Bedaf et al. [42] found that tasks such as self-maintenance activities such as showering, toileting and getting dressed were considered too delicate to be delegated to a robot. In addition, Nyholm et al. [43] found that participants maintained that robots could not in any way be considered caring and that robots were perceived as impersonal and unable to do what human caregivers could do. Furthermore, these findings highlighted the importance of service user participation in innovation and implementation processes. As such, several studies focused on the lack of user participation in innovation processes in the context of digital technologies [44,45,46,47] and suggested that little has been done to create a rigorous and standardized innovation process [48,49]. In the context of robotic technology in the healthcare of people who are over the age of 65, our findings suggest that careful assessment, which includes methods of service user involvement, should be made regarding whether and in what situations it is appropriate to replace human care, and not only human attention, with care robots. Johansson–Pajala & Gustafsson [50] have argued that human encounters could never be replaced, but care robots may complement human interaction. Our findings support this statement.

### Strengths and Limitations

The strength of this systematic review is that the literature search was conducted in collaboration with an experienced academic librarian, and the screening, quality appraisals, data extraction and analysis were conducted in close collaboration between all authors. This increased the validity and reliability of the study.

Nevertheless, this study also had some limitations. Firstly, using a different tool for critical appraisal may have led to some of the included articles being excluded. For example, we noted that participants who did not interact with the robot in two of the included studies were excluded from the analysis [31,51]. In our opinion, this may be seen as a problem. In addition, several of the studies only focused on the positive effects of the robotic technology explored and not on the negative effects [27,29]. Furthermore, Robinson et al. [23] compared very different robotic technologies in relation to people with dementia, such as the guide robot, which is 1.6 m tall, and the seal robot Paro. Such a comparison may not have been appropriate in the context of healthcare provision.

Secondly, it was taken into consideration that participants in the studies included in this current review mainly represented people who were not familiar with everyday technologies. Hence, being exposed to robots that include sound or other visible effects may be sufficient to attract attention. 

Thirdly, we limited the inclusion of articles to those of ‘high’ or ‘medium’ quality. Due to our broad research aim of identifying ‘how robots are utilized in the healthcare of people who are over the age of 65, and how this population experiences interacting with healthcare robots’, we could have conducted a scoping review, which would have allowed for not assessing the quality of the articles. Moreover, our inclusion and inclusion criteria may have excluded articles that could have added interesting information about robotic technology with regard to the care of older people.

## 5. Conclusions

This systematic review indicated that high-quality research on the use of robotic technology in the healthcare of older people above 65 years of age is limited. Hence, there is not enough data to prove that the use of robots for elderly people is a better approach than traditional care. The included studies focused mainly on older people with dementia or cognitive impairment and on social assistive or engagement robots. The usefulness of such robots in a healthcare service under pressure may be discussed. Our findings suggest that the current literature could benefit from research employing longitudinal methods, including the experiences and perspectives of older people, relatives, personnel and stakeholders.

### Implications for Practice

The findings of the current review indicate that there is great potential for further exploration, development and implementation of robotic technology in older peoples’ healthcare in responding to the current and future lack of human resources. Aspects such as dignity, autonomy and social relations should be carefully considered in the processes of implementing robotic technology in the healthcare of older people.

## Figures and Tables

**Figure 1 healthcare-11-00904-f001:**
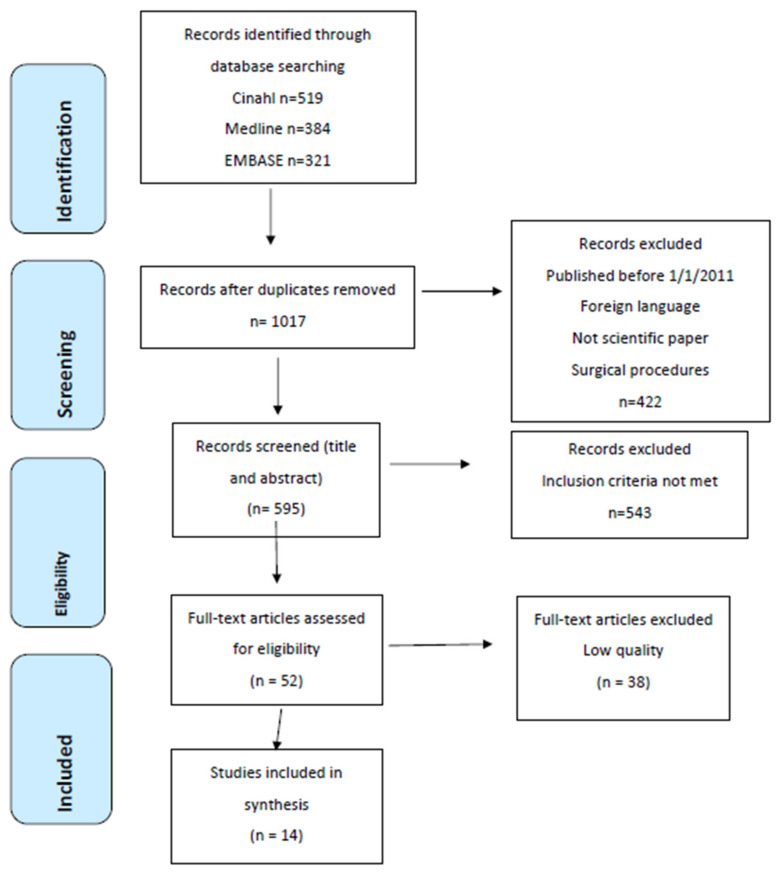
PRISMA 2009 Flow Diagram.

**Table 1 healthcare-11-00904-t001:** PICO framework Use of robots in older people’s healthcare.

P-Patient	I-Intervention	C-Comparison	O-Outcomes
Older Older adults Older persons Geriatric Frail/Frailty Aged Elderly Elder Elderly care Age 65 or above Nursing Homes Health services	Humanoid robot Robotics Artificial intelligence Mobile service robot Assistive robot Artificial social care Service robot Emotional robot Therapeutic robot		

**Table 2 healthcare-11-00904-t002:** Example of search string in Ovid MEDLINE (R).

	Search Words and Boolean Operators	Number of Records
1	Aged/ or “Aged, 80 and over”/ or geriatrics/	3,278,004
2	(old or older or geriatric* or elderly or frail or frailty or geriatric or aged or elder or nursing homes).ti.	426,409
3	Health Services for the Aged/	18,010
4	1 or 2 or 3	3,462,691
5	((humanoid or human or mobile or assistive or intelligent or emotional or therapeutic or service) adj2 (robot or robots or robotic*)).mp. [mp = title, abstract, original title, name of substance word, subject heading word, floating sub-heading word, keyword heading word, organism supplementary concept word, protocol supplementary concept word, rare disease supplementary concept word, unique identifier, synonyms]	3522
6	4 and 5	252
7	(old or older or geriatric* or elderly or frail or frailty or geriatric or aged or elder or nursing homes).ti,ab.	2,174,500
8	1 or 3 or 7	4,791,186
9	5 and 8	384

## Data Availability

Materials are available from the corresponding author upon request.

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
