# Peer review of "The Use of Robotic Technology in the Healthcare of People above the Age of 65—A Systematic Review"

_healthcare, 2023, doi:10.3390/healthcare11060904_

Round 1

Reviewer 1 Report

The article presents an important academic and scientific contribution in a little explored area, such as the review of robotic technologies for people above the age of 65. The systematic review was carried out in detail, generating an interesting discussion on the subject.

There remains some doubt as to the criteria under which the articles investigated were classified as low, medium or high quality (38 articles were excluded due to low quality).

Finally, I leave some minor recommendations on the writing of the document:

- The word "increased" is repeated in lines 27 and 28.

- The sentence "Systematic review 58 registration number: 290621 PROSPERO" in line 58 should be accompanied by a better explanation.

- Remove the word "please" on line 103.

- Carefully unify the formatting in table 3 (punctuation marks, words cut as depress-ion, environment s). Thinking about reducing the font size for this table.

  •  

Author Response

Reviewer 1

Comment 1: The article presents an important academic and scientific contribution in a little explored area, such as the review of robotic technologies for people above the age of 65. The systematic review was carried out in detail, generating an interesting discussion on the subject.

Response: We thank the reviewer for acknowledging our manuscript as important.

Comment 2: There remains some doubt as to the criteria under which the articles investigated were classified as low, medium or high quality (38 articles were excluded due to low quality).

Response: We are truly sorry for this unclarity. We have added som information about the CASP tool scoring system in the revised manuscript: «We used a simple scoring system where ‘criterion is completely met’ = 2, ‘criterion is partially met’ = 1, and ‘criterion not applicable, not met, or not mentioned’ = 0. A total score of 20 = was interpreted as high quality; 16–19 as moderate quality; and ≤ 15 as low quality». Please, see manuscript with track changes.

Comment 3: Finally, I leave some minor recommendations on the writing of the document:

Responses are inserted after each recommendation

- The word "increased" is repeated in lines 27 and 28- Revised

- The sentence "Systematic review 58 registration number: 290621 PROSPERO" in line 58 should be accompanied by a better explanation – An explanation has been included. Please see manuscript with track changes.

- Remove the word "please" on line 103.- Removed

- Carefully unify the formatting in table 3 (punctuation marks, words cut as depress-ion, environment s). Thinking about reducing the font size for this table- The table has been unified. We have put table 3 in a separated document that allows for a wide format.

Reviewer 2 Report

The paper is an interesting recap analysis about robot use in elderly people. However the SR didn’t achieve conclusive results because of the paucity of available data on this issue. Moreover the methodology used by the authors showed some shortcomings especially in the explanation of selection criteria therefore these issues should be addressed. The paper should undergo major revisions before further assessment for publishing.

Methods and materials:

·      Line 87: Among inclusion criteria there is “full text available”, so how many are excluded due to this lack? Not specified (from line 101 to line 116)

·      Line 118: Please, describe CASP Criteria with more details (Randomized studies and Case Control Group studies are included?)

·      Line 124: Please, explain why 38 studies are low in quality with CASP criteria

Results:

·      Table 3: format to make table more user friendly even the amount information should be reduced

·      Lines 188-193: Koshla et al. study has a sample of 5 older people: how is it possible that the CASP checklist considers it to be a high quality study?

·      Line 187: “periode” without “e”

·      Line 188: Further information necessary to explain (n=2043) number of observations related to 5 patients in the study

·      Line 132: missing open bracket (?)

Conclusion:

·      Line 358: “peope” missing “l”

·      Please, add that there is not enough data to prove that the use of robots in elderly people is a better approach because large-scale benefits are lacking, so codified studies should be conducted.

Author Response

Reviewer 2

Comment 1: The paper is an interesting recap analysis about robot use in elderly people. However the SR didn’t achieve conclusive results because of the paucity of available data on this issue. Moreover the methodology used by the authors showed some shortcomings especially in the explanation of selection criteria therefore these issues should be addressed. The paper should undergo major revisions before further assessment for publishing.

Response: We thank the reviewer for acknowledging our manuscript as interesting,   and for pointing out the shortcomings, and allowing us to revise accordingly.

Methods and materials

Comment 2: Line 87: Among inclusion criteria there is “full text available”, so how many are excluded due to this lack? Not specified (from line 101 to line 116)

Response: We thank the reviewer for making us aware of this. We did not use «full-text available» as an inclusion criteria. Hence, no papers were excluded due to not being available in full-text. We have revised the inclusion criteria accordingly.

Comment 3: Line 118: Please, describe CASP Criteria with more details (Randomized studies and Case Control Group studies are included?)

Response: We thank the reviewer for requesting this. We have added more information about the CASP in the revised mansucript. Please, see manuscript with track changes.

Comment 4: Line 124: Please, explain why 38 studies are low in quality with CASP criteria

Response:Please, see response to reviewer 1, comment 2, as well as above. Please, see page 6 in the manuscript with track changes.

Results:

Comment 5: Table 3: format to make table more user friendly even the amount information should be reduced

Response: We thank for this input, and have formatted the table to be more reader friendly. We hope our revisions are deemed sufficient- otherwise, we will of course revise again.

Comment 6: Lines 188-193: Koshla et al. study has a sample of 5 older people: how is it possible that the CASP checklist considers it to be a high quality study?

Response: The reviewer poses an interesting question. However, the CASP cheklist does not evaluate the sample size. In line with a qualitative approach, the trustworthiness of the results is based in the depth and richness. This study was based on observations of the five participants, over three months. Please also see the response to comment 7.

Comment 6: Line 187: “periode” without “e”- Revised

Comment 7: Line 188: Further information necessary to explain (n=2043) number of observations related to 5 patients in the study

Response: The patients were observed, and video-recorded through the robots, over a three months period. We have added this information in the revised manuscript: «Further, Khosla et al. (27) performed observations (n=2043) of older people with dementia interacting with Betty, through video recording through the robot in their homes over a three months period».

Comment 8: Line 132: missing open bracket (?)

Response: Thank you. We have thoroughly proof-read the manuscript.

Conclusion:

Comment 9: Line 358: “peope” missing “l”- Revised

Comment 10: Please, add that there is not enough data to prove that the use of robots in elderly people is a better approach because large-scale benefits are lacking, so codified studies should be conducted.

Response: We thank the reviewer for this suggestion, and have consequently added to the conclusion. Please, see manuscript with track changes.

Reviewer 3 Report

The systematic review provides a timely overview of the filed of robotics in the health care of people above the age of 65. The mansucript is well-structured and well-written. In the attached pdf-file, I have added a few comments that hopefully contribute to improving the quality of the manuscript.

A major issue with the manuscript is that the aim of the review is not fully reflected among the search terms. More specifically, the aim (and title) include social services, but social services is not mentioned among the search words nor in the PICO framework. This needs to be addressed either by rephrasing the aim and title, or conducting a complementary search, or explaining and justifying the reason for the discrepancy.

A medium issue that needs to be addressed is the presentation of the search string in table 2. It appears that table 2 includes unnecessary information, i.e., searches that do not impact the final results.

The remaining issues are all minor.

Author Response

Reviewer 3

Comment 1: The systematic review provides a timely overview of the filed of robotics in the health care of people above the age of 65. The manuscript is well-structured and well-written. In the attached pdf-file, I have added a few comments that hopefully contribute to improving the quality of the manuscript.

Response: We thank the reviewer for acknowledging our manuscript as timely, well-structured and well-written. We also thank for the comments, which we mean have contributed to improving the manuscript.

Comment 2: A major issue with the manuscript is that the aim of the review is not fully reflected among the search terms. More specifically, the aim (and title) include social services, but social services is not mentioned among the search words nor in the PICO framework. This needs to be addressed either by rephrasing the aim and title, or conducting a complementary search, or explaining and justifying the reason for the discrepancy.

Response: We thank the reviewer for this timely comment. We have chosen to rephrase the aim and title, so that these adhere to the literature search.

Comment 3: A medium issue that needs to be addressed is the presentation of the search string in table 2. It appears that table 2 includes unnecessary information, i.e., searches that do not impact the final results.

Response: We thank the reviewer for this input. However, table 2 shows an example of search string in one of the databases. This has be specified in the table heading. Moreover, the search presents the combination of ‘OR’ and ‘AND’ searches, which we mean is meaningful. If the reviewer disagree, we will of course revise accordingly

Round 2

Reviewer 2 Report

Dear authors,

thank you for responding to all comments. I believe that your work can be published. 

The revised table is really improved, but I think that a shortened version should be included in the main paper. The shortened version should include: first author, title, year publishing, sample size and main objective of the paper.

With my best regards

Author Response

Thank you for acknowledging our revisions. We have followed your suggestion regarding the table, but have also included a column describing the technology used. This is due to referring to this in text, and to provide the readers with background information about the different robots. We hope this is assumed acceptable. 

Reviewer 3 Report

Congratulations on the improved mansucript. Although I think it is superfluous  and potentially misguiding to present searches that do not affect the final search in a database, it is possible to keep it as an example, even if it is ineffective.

A handful of "social care" remain in the text, including in the citation. Please revise where necessary.

Author Response

We thank you for acknowledging our revisions. We have of course removed the ‘social care’ in text- where appropriate (not included in e.g. results/titles etc) 
